# Total Oxidation of Toluene and Propane over Co$_3$O$_4$ Catalysts: Influence of Precipitating pH and Washing

Imane Driouch [1] , Weidong Zhang [1], Michèle Heitz [2], Jose Luis Valverde [3] and Anne Giroir-Fendler [1,*]

1   Univ. Lyon, Université Claude Bernard Lyon 1, CNRS, IRCELYON, 2 Avenue Albert Einstein, F-69622 Villeurbanne, France; imane.driouch@etu.univ-lyon1.fr (I.D.); weidong.zhang@ircelyon.univ-lyon1.fr (W.Z.)
2   Department of Chemical Engineering and Biotechnological Engineering, Faculty of Engineering, Université de Sherbrooke, 2500 Boul. de l'Université, Sherbrooke, QC J1K 2R1, Canada; michele.heitz@USherbrooke.ca
3   Department of Chemical Engineering, Faculty of Chemical Science and Technology, University of Castilla-La Mancha, Avenida Camilo José Cela 12, 13005 Ciudad Real, Spain; jlvalverde1964@gmail.com
*   Correspondence: anne.giroir-fendler@ircelyon.univ-lyon1.fr; Tel.: +33-472-431-586

**Abstract:** A series of Co$_3$O$_4$ catalysts were synthesized by an ammonia precipitation method at various precipitating pH values (8.0, 8.5, 9.0, 9.5, and 10.0) and with different numbers of washings. Their performance in the total oxidation of two selected hydrocarbons, toluene and propane, was evaluated at a reactant/oxygen molar ratio of 1/210 and a Weight Hourly Space Velocity (WHSV) of 40,000 mL g$^{-1}$ h$^{-1}$. The physicochemical properties of the catalysts were characterized by thermogravimetric and differential thermal analysis (TG/DTA), Fourier-transform infrared spectroscopy (FTIR), X-ray diffraction (XRD) and N$_2$ absorption–desorption. The results show that the catalysts are in the cubic spinel phase (Fd-3m (227), a = 8.0840 Å) with average crystalline sizes of 29−40 nm and specific surface areas of 12–20 m$^2$ g$^{-1}$. All catalysts allowed 100% conversion of both toluene and propane at temperatures below 350 °C. The precipitating pH and the number of washings were observed to significantly affect the catalytic performance. The optimal synthesis condition was established to be pH 8.5 with two washings. The best catalyst gave 100% conversion of toluene and propane at 306 °C and 268 °C, respectively.

**Keywords:** cobalt oxide; effect of pH; effect of washing; toluene oxidation; propane oxidation

## 1. Introduction

Volatile organic compounds (VOCs) represent one of the major issues of this century because of their participation in atmospheric photochemical reactions, since they cause an increase of ozone (O$_3$) concentration in the troposphere, produce photochemical smog, and sometimes form tiny health-damaging particulate matter (PM) [1]. In fact, ozone causes respiratory and cardiovascular diseases, whereas PM causes pulmonary diseases. Recent research also shows that polluted air by O$_3$ and PM had an impact on central nervous system diseases, including Alzheimer's and Parkinson's ones, and strokes [2]. For these reasons, many researchers are focusing on finding the best way to eliminate VOCs. Catalytic oxidation, one of the most commonly used methods for VOC abatement, with the advantages of high-efficiency, energy-saving and less or non-secondary pollutant, has attracted attention in recent decades. Supported noble metal (Pd, Pt, Ru, Rh, Au, etc.) catalysts are generally used for the total oxidation of VOCs. However, some drawbacks such as their high price, low availability, volatility and sintering issue, and susceptibly poisoning tendency, limit their application. Sihaib et al. studied the effect of citric acid concentration on the activity of LaMnO$_3$ catalyst and found that the most active LaMnO$_3$ exhibited catalytic performance comparable to that of Pd/Al$_2$O$_3$ catalysts [3].

Liu et al. prepared a highly active and moisture-resistant $MnO_2$-based catalyst for low-temperature benzene removal by a redox method and nitric acid post-processing [4]. Tang et al. boosted the activity of $Co_3O_4$ by an easy acetic acid etching method and obtained a novel $Co_3O_4$ catalyst with much higher activity than either commercial Pt catalyst or a $Pd/Al_2O_3$ one [5]. Among various non-noble catalysts, $Co_3O_4$ was established to be one of the most active, especially for carbon monoxide (CO) and hydrocarbons oxidation [6–9].

With the aim of further reducing energy consumption, many efforts have been made to combust VOCs over $Co_3O_4$ catalysts at lower temperatures. One effective way is to use porous material by the hard or soft template method since high porosity favours mass transfer and adsorption [10,11]. Another way to improve catalytic performance is to dope catalysts with metal or mixed oxides [12,13]. In this way, oxygen mobility and reducibility might be enhanced, and the synergistic effect could be generated [14–16]. Moreover, it is also possible, by tuning the synthesis parameters, such as ageing time [17], precipitating pH [18] and calcination temperature [19], to achieve higher catalytic activity.

Precipitation or co-precipitation methods are widely used for cobalt oxide catalyst synthesis due to simple operation, mild condition, and the possibility of mass production. Precipitation agent plays a very important role in this method. $Na_2CO_3$, NaOH, and ammonia are the most used alkaline precipitants. However, it has been reported that residual $Na^+$ in $Co_3O_4$ negatively affected the catalytic activity of methane oxidation [20]. Similarly, Tang et al. found that alkali-metal (Li, Na, K) had a poisoning effect on the oxidation of both CO and propane over $Co_3O_4$ catalysts [21]. Given this fact, the ammonia route was preferable because the $NH_4NO_3$ species would completely decompose after calcination, ruling out the effect of impurities issues on catalytic activity. In the case of the reaction between the cobalt salt and ammonia, a precipitation–complexing competition balance exists. Thus, $Co^{2+}$ reacts with $OH^-$ to produce $Co(OH)_2$; the as-formed $Co(OH)_2$ then reacts with free $NH_3$ to generate $Co(NH_3)_6^{2+}$. This reaction equilibrium is dependent on the pH and the amount of free $NH_3$ in the solution. Deng successfully prepared NiO by the ammonia precipitation method and pointed out the importance of the pH control [22]. In addition, Muhamad et al. reported that increasing the number of washings after ammonia precipitating positively affected the surface area of CuO [23]. To the best of our knowledge, there is a lack of information in the literature about the $Co_3O_4$ synthesis by ammonia route at different pH values and number of washings.

Herein, a series of $Co_3O_4$-based catalysts were prepared via the ammonia-precipitation method at different pH values and with different number of washings, characterized by thermogravimetric and differential thermal analysis (TG/DTA), Fourier-transform infrared spectroscopy (FTIR), and X-ray diffraction (XRD), and evaluated for the total oxidation of toluene and propane.

## 2. Results and Discussion

### 2.1. Influence of the Precipitating pH

#### 2.1.1. TG/DTA, FTIR, XRD and $N_2$ Adsorption Characterizations

With the aim to clarify the decomposition process of the cobalt precursor and the thermal stability of the formed cobalt oxide, TG/DTA analysis was conducted and the curves of cobalt precursors prepared at different pH values are plotted in Figure 1. For all samples, the TG graph showed two similar decomposition stages. The first stage from 25 °C to 150 °C with a mild weight loss of 4.2 wt.% could be ascribed to the removal of hydrated water. The second stage between 150 and 300 °C exhibited a total weight loss ranging from 34 to 37 wt.%, which was much larger than that expected for the decomposition of cobalt hydroxide into cobalt oxide (13.6 wt.%). Therefore, there must be some $NH_4NO_3$ or $[Co(NH_3)_6]^{2+}$ incorporated in the precursor; much of the weight loss was due to the elimination of these species. DTA profiles showed two exothermic peaks in the range of 150−300 °C, corresponding to the phase transition stage. For the transformation of pure cobalt hydroxide into cobalt oxide, an endothermic decomposition peak followed by an exothermic oxidation peak was expected. However, the decomposition and oxidation of $NH_4NO_3$ or $[Co(NH_3)_6]^{2+}$ should release a lot

of heat, confirming the presence of the impurity in the cobalt precursor. From 300 °C to 500 °C, only 1% of weight loss was observed, suggesting the thermal stability of cobalt oxide. The final calcination temperature in this work was set at 500 °C.

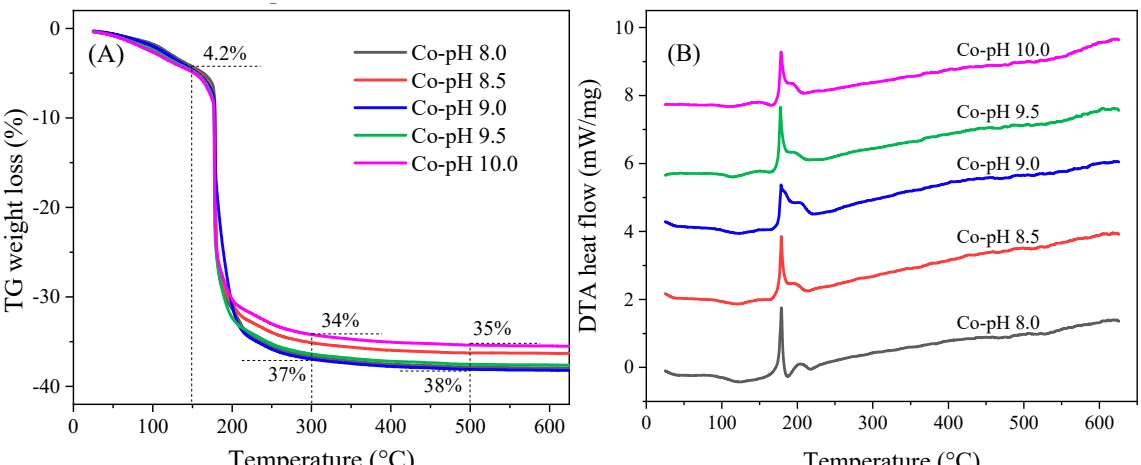

**Figure 1.** (**A**) thermogravimetric (TG) and (**B**) differential thermal analysis (DTA) curves of cobalt precursors prepared at different pH values.

Figure 2 shows the FTIR spectra of cobalt precursors and cobalt oxides prepared at different pH values. Regarding the spectra of the 80 °C-dried samples, the broad bands at 3620, 3230 and 3050 cm$^{-1}$ were ascribed to the characteristic stretching vibration modes of $NH_3$ group [24]. The cobalt ammonia complex formation was evidenced by the bands at 1750 cm$^{-1}$ and 1628 cm$^{-1}$ [24,25]. The bands at 1470, 1308 and 828 cm$^{-1}$ were assigned to the symmetric and asymmetric stretching vibrations of $NO_3^-$ species [26]. The bands at 990 and 510 cm$^{-1}$, together with the band at 626 cm$^{-1}$, originated from the Co–OH bending and Co–O stretching vibrations, respectively [26–28]. After calcination, only two intense bands are observed in Figure 2B. The band at 540 cm$^{-1}$ was associated with the stretching vibration of O–$(Co^{3+})_3$ where $Co^{3+}$ was in the octahedral hole in the spinel lattice, and the other band at 655 cm$^{-1}$ was attributed to the stretching vibration of $Co^{2+}$–$Co^{3+}$–$O_3$ where $Co^{2+}$ was in the tetrahedral hole [29,30]. The occurrence of fingerprint bands of $Co_3O_4$ and the absence of impurity band would demonstrate the fully development of $Co_3O_4$ spinel after annealing at 500 °C, which was coincidental with the TG/DTA findings.

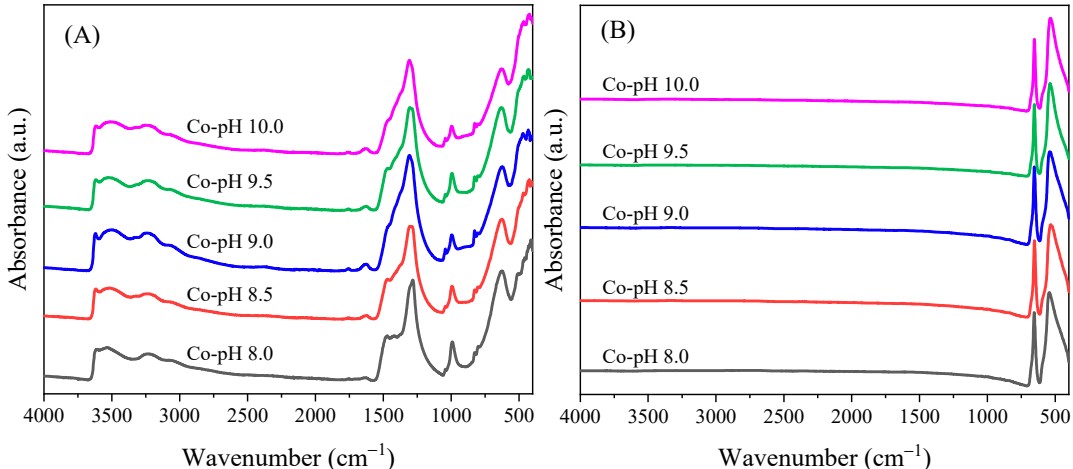

**Figure 2.** Fourier transform-infrared spectroscopy (FTIR) spectra of (**A**) cobalt precursors and (**B**) cobalt oxides prepared at different pH values.

The mass of collected $Co_3O_4$ after annealing was weighed and divided by the theoretical mass of $Co_3O_4$ assuming that all Co was converted into $Co_3O_4$. The estimated product yields were summarized in Table 1. As expected, low pH cannot ensure the complete precipitation of $Co^{2+}$ while high $NH_3$ concentration would induce the dissolution of $Co(OH)_2$ and subsequent transformation into $Co(NH_3)_6{}^{2+}$. With a precipitating pH value of 9, the highest yield was achieved: 92%.

**Table 1.** Product yields, crystalline sizes, lattice constants and textural data of cobalt oxides prepared at different pH values.

| Catalysts | Yield (%) [a] | d (nm) [b] | a (Å) [b] | SSA ($m^2 g^{-1}$) [c] | $V_p$ ($cm^3 g^{-1}$) [c] | $D_p$ (nm) [c] |
|---|---|---|---|---|---|---|
| Co-pH 8.0 | 64 | 36 | 8.0838 | 12 | 0.052 | 18 |
| Co-pH 8.5 | 87 | 40 | 8.0853 | 14 | 0.035 | 9 |
| Co-pH 9.0 | 92 | 37 | 8.0853 | 13 | 0.046 | 15 |
| Co-pH 9.5 | 89 | 33 | 8.0841 | 12 | 0.052 | 17 |
| Co-pH 10.0 | 87 | 32 | 8.0830 | 16 | 0.055 | 14 |

[a] Product yields estimated by assuming that all Co was converted into $Co_3O_4$. [b] Average crystalline sizes and lattice constants calculated from XRD patterns. [c] Specific surface areas, total pore volumes and average pore sizes obtained from $N_2$ adsorption isotherms.

The XRD patterns of cobalt oxides prepared at different pH values are presented in Figure 3. All samples exhibited well-defined diffraction peaks at 19.0°, 31.3°, 36.8°, 38.5°, 44.8°, 55.6°, 59.4° and 65.2°, matching well with the (111), (220), (311), (222), (400), (422), (511) and (440) planes of cubic spinel $Co_3O_4$ (JCPDS PDF # 74–2102, Fd-3m (227), a = 8.0840 Å). No peak corresponding to CoO or other impurities were observed, indicating the high purity of the $Co_3O_4$ product. By using the Jade software, the average crystalline sizes (d) and lattice constants (a) were calculated based on the six strongest planes (111), (220), (311), (400), (511) and (440). Data are listed in Table 1. The average crystalline sizes are in the range of 32–40 nm, with Co-pH 8.5 and Co-pH 10.0 showing the largest and the smallest values, respectively. No correlation trend with the precipitating pH was found. The lattice constants were close to that of the standard PDF card, showing no obvious difference among each sample.

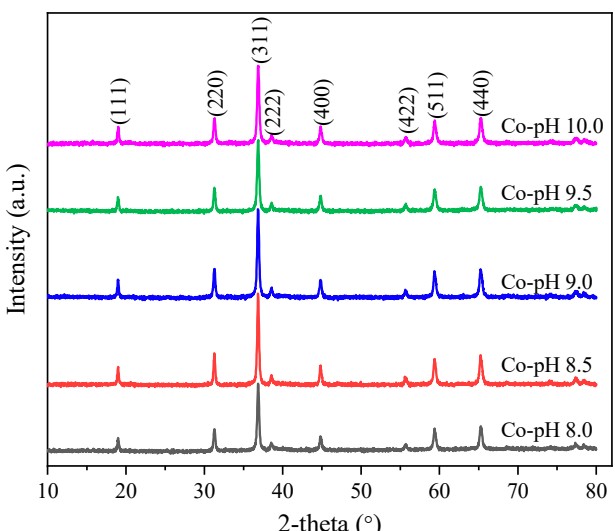

**Figure 3.** X-ray diffraction (XRD) patterns of cobalt oxides prepared at different pH values.

Figure 4 displays the adsorption–desorption isotherms and pore size distributions of the cobalt oxide-based catalysts prepared at different pH values. As it can be seen, the shapes of these isotherms were similar and correspond to the type IV isotherms with a small hysteresis loop, indicating the presence of accumulated mesopores. The pore size distributions of all samples were broad and irregular, due to the random accumulation of $Co_3O_4$ nanoparticles. The textural properties in terms of specific

surface area (SSA), total pore volume ($V_p$) and average pore size ($D_p$) are listed in Table 1. The SSA values were between 12.2 and 15.5 $m^2$ $g^{-1}$. Co-pH 10.0 possessed the largest SSA, which matched well with the smallest crystalline size obtained from the XRD analysis. However, Co-pH 8.5 also exhibited a relatively large SSA (14.4 $m^2$ $g^{-1}$) though its crystalline size was the biggest. This might mean that the aggregation extent of Co-pH 8.5 nanoparticles was less severe. The total pore volumes of all samples were in the range of 0.035–0.055 $m^3$ $g^{-1}$.

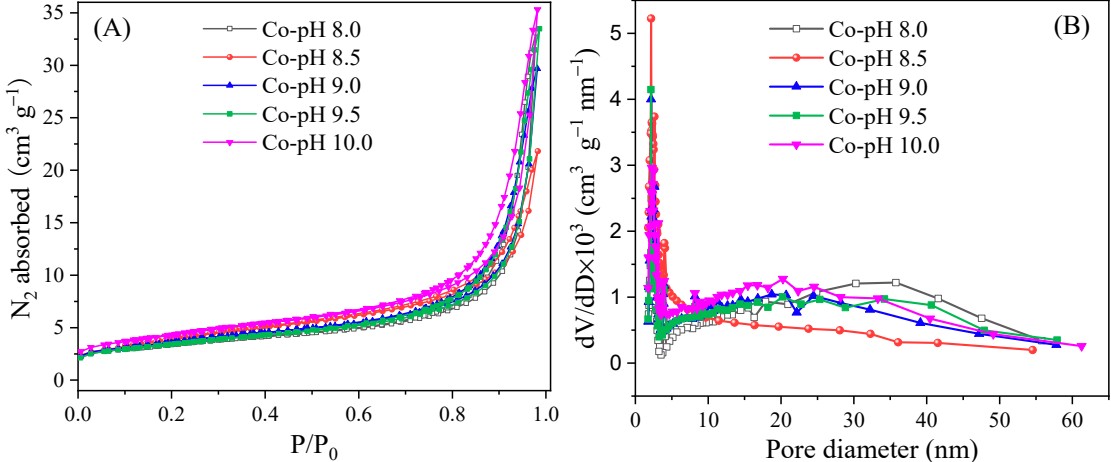

**Figure 4.** (**A**) $N_2$ adsorption−desorption isotherms and (**B**) pore size distributions of cobalt oxides prepared at different pH values.

In summary, no evident relationship was found between the structure/porosity of the $Co_3O_4$ catalysts and the pH of the precipitating process. According to the TG and FTIR results, a large and uncontrollable amount of impurities ($NH_4^+$, $NO_3^-$, $Co(NH_3)_6^{2+}$, etc.) was maintained in the surface or incorporated into the interlayer of the $Co(OH)_2$ precursors, which may affect each catalysts differently during the calcination process, accounting for this phenomenon.

### 2.1.2. Catalytic Performance in the Toluene and Propane Oxidation

All catalysts were tested for the total oxidation of toluene and propane, and the light-off curves are shown in Figure 5, whereas the temperature values for achieving 10%, 50% and 90% conversions ($T_{10}$, $T_{50}$ and $T_{90}$) are listed in Table 2. Regarding the toluene oxidation, the cobalt oxide synthesized at pH = 9.0 presented the best catalytic performance, with $T_{90}$ of 282 °C, followed by Co-pH 8.5 and Co-pH 8.0 showing similar performance. The worst catalyst in terms of performance was Co-pH 9.5, with $T_{90}$ of 299 °C. The $T_{90}$ of the optimal catalyst Co-pH 9.0 was 17 °C lower than that the temperature of the worst catalyst Co-pH 9.5, demonstrating the effect of precipitating pH on the activity of the final catalyst. Regarding the propane oxidation, the most efficient catalyst was Co-pH 8.5 while Co-pH 10.0 was the less reactive one. The difference between the $T_{90}$ of these two catalysts was 31 °C, which would demonstrate the influence of the value of the precipitating pH. The sequence of reactivity for propane oxidation was the following: Co-pH 8.5 > Co-pH 8.0 > Co-pH 9.0 > Co-pH 9.5 > Co-pH 10.0.

The cycle stability of these $Co_3O_4$-based catalysts was checked using Co-pH 9.0 as the representative one. As shown in Figure S1, there is no distinction among the three light-off curves during cooling run for both toluene and propane oxidation, proving the excellent cycle stability of $Co_3O_4$ catalysts.

The differences in performance of the different catalysts in the oxidation of toluene were not so pronounced as that in propane oxidation. Finally, $Co_3O_4$ catalyst synthesized at pH of 8.5 was regarded as the best candidate for VOCs removal. As a result, a precipitating pH of 8.5 was used in the following study.

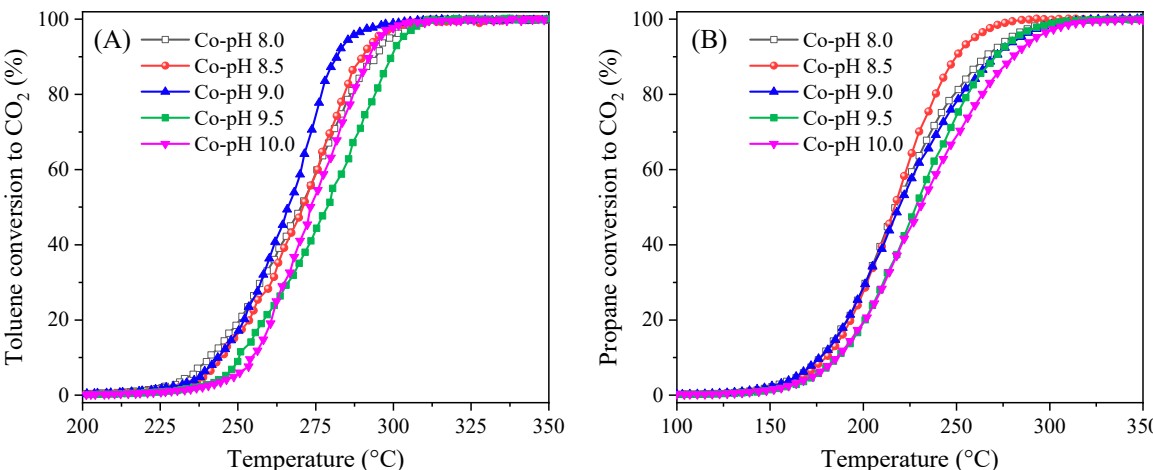

**Figure 5.** (**A**) Toluene and (**B**) propane oxidation over cobalt oxides prepared at different pH values.

**Table 2.** Temperature values of 10% ($T_{10}$), 50% ($T_{50}$) and 90% ($T_{90}$) toluene and propane conversion of cobalt oxides prepared at different pH values.

| Catalysts | Toluene Oxidation | | | Propane Oxidation | | |
|---|---|---|---|---|---|---|
| | $T_{10}$ | $T_{50}$ | $T_{90}$ | $T_{10}$ | $T_{50}$ | $T_{90}$ |
| Co-pH 8.0 | 240 | 270 | 294 | 177 | 217 | 265 |
| Co-pH 8.5 | 244 | 271 | 290 | 180 | 216 | 249 |
| Co-pH 9.0 | 243 | 266 | 282 | 176 | 219 | 270 |
| Co-pH 9.5 | 250 | 273 | 299 | 186 | 228 | 270 |
| Co-pH 10.0 | 254 | 278 | 292 | 186 | 230 | 280 |

## 2.2. Influence of the Number of Washings

### 2.2.1. TG/DTA, FTIR, XRD and $N_2$ Adsorption Characterizations

The TG/DTA curves displayed in Figure 6 show a totally different decomposition behavior between the unwashed sample and the washed samples. For the unwashed sample, a weight loss of 4.5% due to dehydration was observed, while just a weight loss of 0.6% occurred for the sample washed once. Moreover, when the temperature raised from 150 to 300 °C, a weight loss of 35% can be observed for the unwashed sample, which can be attributed to a complex series of overlapping reactions including denitration, decomposition of cobalt ammonia complex, phase transition of cobalt hydroxide, oxidation of $Co^{2+}$ accompanying the formation of $Co_3O_4$, and combustion of the evolved gases [31]. This process was accompanied by two exothermic peaks in the DTA plot. However, in the case of the sample washed once, only half of weight loss (18%) along with an endothermic peak centered at 197 °C, was observed. After washing, most $NO_3^-$ and $Co(NH_3)_6^{2+}$ species were removed from the surface of the precursors (see FTIR analysis below). The oxidation of $NO_3^-$ releases large heat whereas the decomposition of $Co(NH_3)_6^{2+}$ and $Co(OH)_2$ is an endothermic process. Therefore, endothermic peaks, rather than exothermic peaks, emerged for the washed samples. With an increase in the number of washings, the difference in weight loss and the size of the endothermic peak were progressively reduced. The slight weight loss (1%) between 300 and 500 °C could be related to the gradual loss of excess oxygen in the initially formed non-stoichiometric $Co_3O_4$ [32].

Figure 7 shows the FTIR spectra of the dried and calcined cobalt samples precipitated at pH 8.5 and using different number of washings. Regarding the spectra of the dried sample, visible differences could be found according the number of washings. When washed once, the bands at 3230 and 3050 cm$^{-1}$ related to the stretching vibrations of $NH_3$ group practically disappeared [24]. Other bands corresponding to $Co(NH_3)_6^{2+}$ and $NO_3^-$ species clearly diminished. Some tiny new bands at 2320–2370 cm$^{-1}$ were associated with atmospheric $CO_2$ [24]. On the other hand, no appreciable

differences were observed in the spectra regardless of the number of washings considered. The same was observed when different cobalt precursors in nature were used. In this case, the spectra exhibited two absorption bands at 540 ($\nu$1) and 655($\nu$2) cm$^{-1}$ corresponding to the $Co_3O_4$ spinel lattice [29].

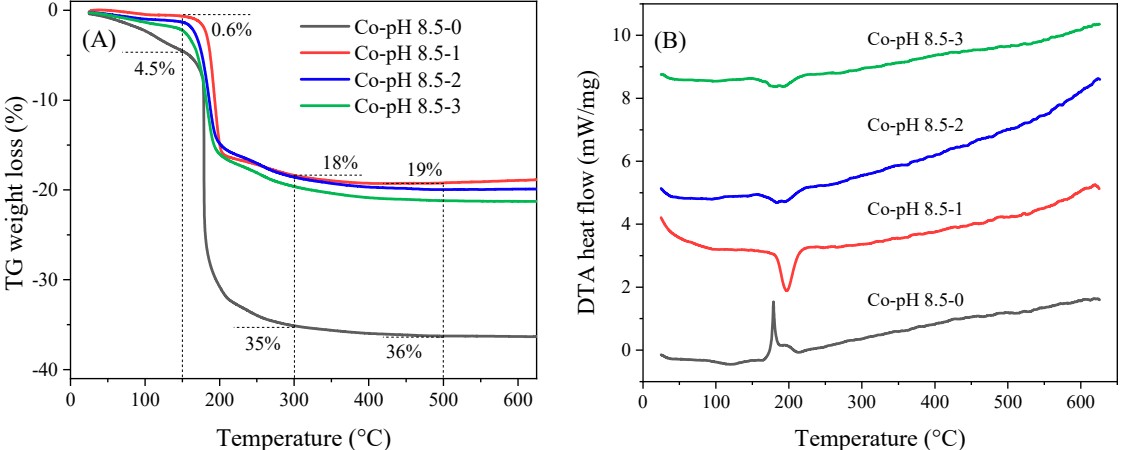

**Figure 6.** (**A**) TG and (**B**) DTA curves of cobalt precursors prepared at pH 8.5 with different number of washings.

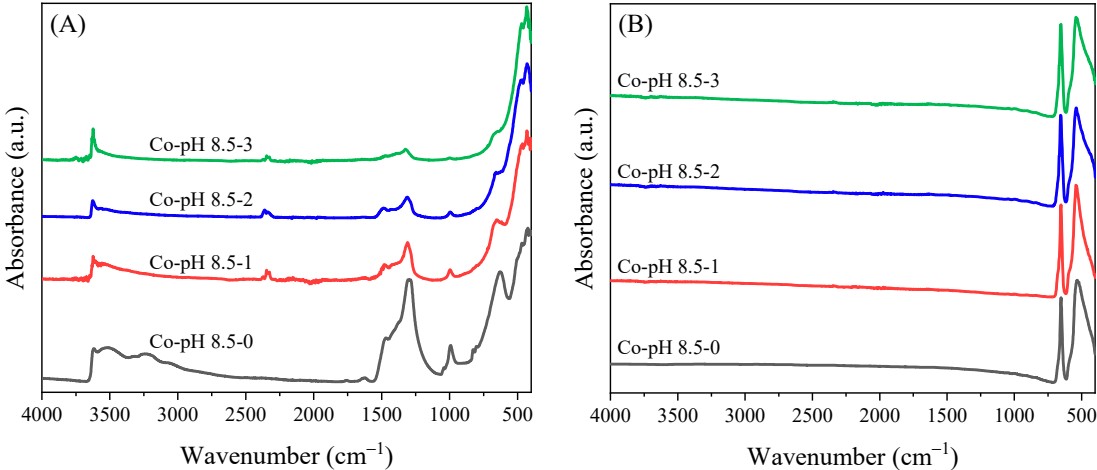

**Figure 7.** FTIR spectra of (**A**) cobalt precursors and (**B**) cobalt oxides prepared at pH 8.5 with different number of washings.

The calculated $Co_3O_4$ yields (Table 3) diminished as the number of washings increased. As expected, the higher the number of washings, the lower the value of pH which would cause the redissolution of the $Co(OH)_2$ precipitate. When three washings were used, this yield fell to 69%.

**Table 3.** Product yields, crystalline sizes, lattice constants and textural of cobalt oxides prepared at pH 8.5 with different number of washings.

| Catalysts | Yield (%) [a] | d (nm) [b] | a (Å) [b] | SSA (m$^2$ g$^{-1}$) [c] | V$_p$ (cm$^3$ g$^{-1}$) [c] | D$_p$ (nm) [c] |
|---|---|---|---|---|---|---|
| Co-pH 8.5-0 | 87 | 40 | 8.0853 | 14 | 0.035 | 9 |
| Co-pH 8.5-1 | 77 | 30 | 8.0882 | 20 | 0.040 | 8 |
| Co-pH 8.5-2 | 73 | 29 | 8.0870 | 19 | 0.039 | 8 |
| Co-pH 8.5-3 | 69 | 32 | 8.0883 | 18 | 0.035 | 8 |

[a] Product yields estimated by assuming that all Co was converted into $Co_3O_4$. [b] Average crystalline sizes and lattice constants calculated from XRD patterns. [c] Specific surface areas, total pore volumes and average pore sizes obtained from N$_2$ adsorption isotherms.

The well-resolved XRD patterns shown in Figure 8 further confirm the formation of the $Co_3O_4$ spinel. The $Co_3O_4$ diffraction lines show that all washed samples presented a much weaker profile than the unwashed one. The average crystallite sizes and the lattice constants estimated by the Jade software reported in Table 3 reveal smaller crystallite sizes and bigger lattice constants for Co-pH 8.5-1, Co-pH 8.5-2 and Co-pH 8.5-3 as compared to those of Co-pH 8.5-0. Similar to the conclusions derived from the FTIR analysis, no influence of the number of washings in the XRD patterns was observed.

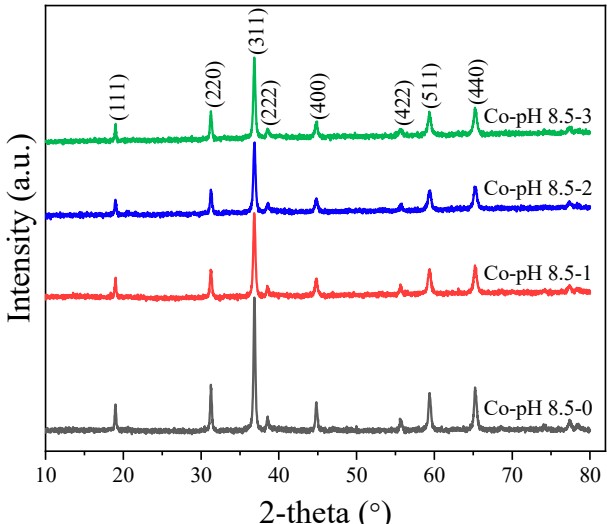

**Figure 8.** XRD patterns of cobalt oxides prepared at pH 8.5 with different number of washings.

Figure 9 provides the $N_2$ adsorption−desorption isotherms and pore size distributions of cobalt oxide catalysts obtained from different numbers of washings. Co-pH 8.5-1 exhibited the highest $N_2$ uptake. Type IV isotherms with small hysteresis loops and similar pore size distributions were again observed for all samples. SSA increased from 14.4 to 20.2 $m^2$ $g^{-1}$ and then decreased a little bit with the increasing number of washings (Table 3). The catalyst washed once had the largest surface area and pore volume.

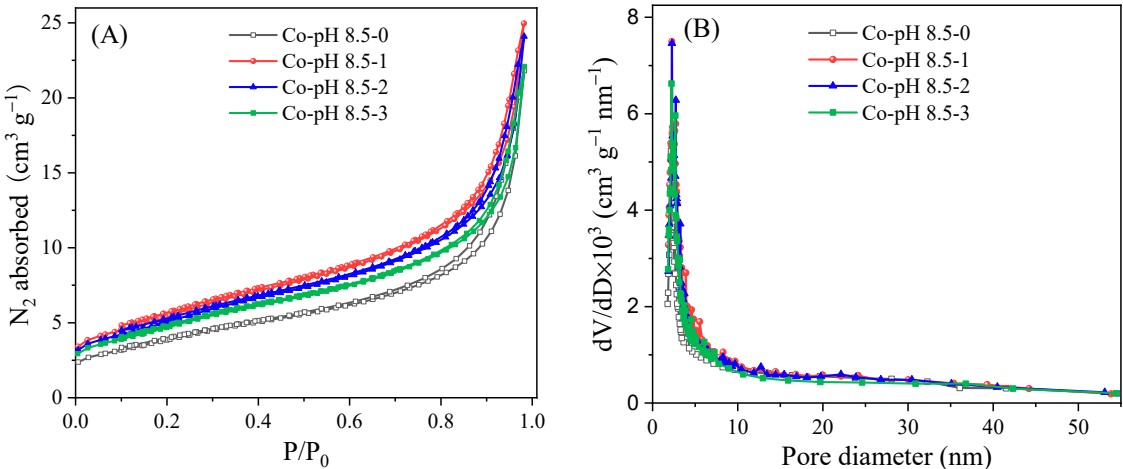

**Figure 9.** (**A**) $N_2$ adsorption−desorption isotherms and (**B**) pore size distributions of cobalt oxides prepared at pH 8.5 with different number of washings.

### 2.2.2. Catalytic Performance in the Toluene and Propane Oxidation

The catalytic performance of all catalysts in the total oxidation of toluene and propane is presented in Figure 10 and Table 4. By comparing the light-off curves and $T_{10}$, $T_{50}$, $T_{90}$ values, $Co_3O_4$ precipitated

at pH = 8.5 and washed twice outperformed other catalysts for both toluene and especially propane oxidation. Regarding Co-pH 8.5-2, toluene conversion started at about 236 °C ($T_{10}$) and reached 90% conversion at 287 °C whereas for propane oxidation, it exhibited a $T_{50}$ of 204 °C, 12 °C lower than that observed for Co-pH 8.5-0.

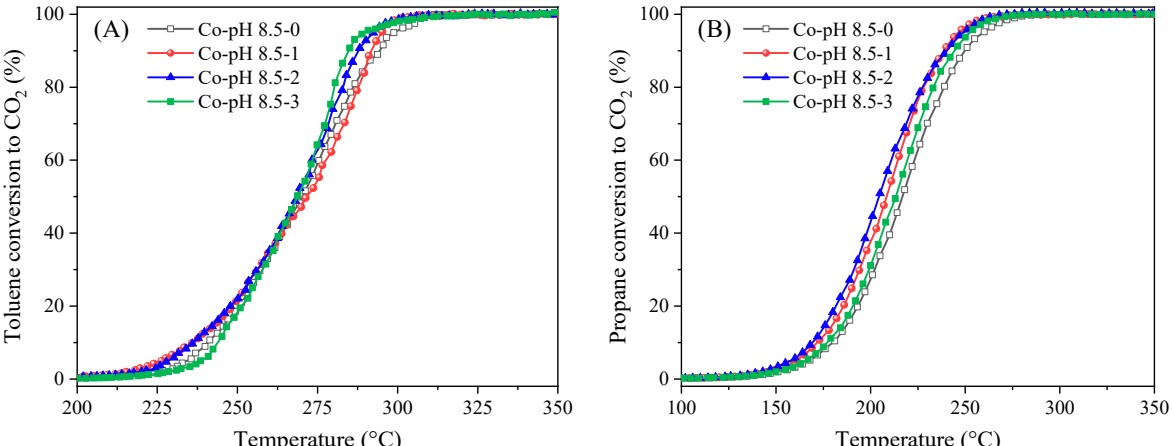

**Figure 10.** (**A**) Toluene and (**B**) propane oxidation over cobalt oxides prepared at pH 8.5 with different number of washings.

**Table 4.** Temperature values of 10% ($T_{10}$), 50% ($T_{50}$) and 90% ($T_{90}$) toluene and propane conversion of cobalt oxides prepared at pH 8.5 with different number of washing.

| Catalysts | Toluene Oxidation | | | Propane Oxidation | | |
|---|---|---|---|---|---|---|
| | $T_{10}$ | $T_{50}$ | $T_{90}$ | $T_{10}$ | $T_{50}$ | $T_{90}$ |
| **Co-pH 8.5-0** | 244 | 271 | 290 | 180 | 216 | 249 |
| **Co-pH 8.5-1** | 236 | 271 | 291 | 171 | 208 | 238 |
| **Co-pH 8.5-2** | 236 | 268 | 287 | 169 | 204 | 240 |
| **Co-pH 8.5-3** | 244 | 268 | 284 | 176 | 213 | 244 |

It is remarkable that all washed catalysts had better performance than the unwashed one. Co-pH 8.5-0 was the less active catalyst in both reactions possibly due to its smaller SSA. This underlined the importance of the washing process after precipitation even if the impurities are decomposed after calcination.

To sum up, the washing step did affect the nature of cobalt precursor and led to enhanced catalytic performance. Considering that an excessive number of washings took more time and energy and resulted in a decline in $Co_3O_4$ yield, the sample washed once was preferred since the activity difference, if compared to that of the sample washed twice, considered as the best catalyst, was not so significant.

### 2.3. Influence of Ageing Time and Precipitating Temperature

Apart from precipitating pH and washing, other factors such as ageing time and precipitating temperature could also have impact on the cobalt precursor and the final product.

Figure S2 and Table S1 show that longer ageing times contributed to the synthesis of porous $Co_3O_4$ material with larger SSA. On the contrary, 80 °C-precipitating yielded a material with smaller SSA. In addition, both treatments would lead to lower $Co_3O_4$ yields.

By comparing the light-off curves and $T_{10}$, $T_{50}$, $T_{90}$ values for toluene and propane oxidation of cobalt oxide catalysts prepared at different conditions (Figure 11 and Table 5), the effect of these two treatments on the catalytic behaviour of $Co_3O_4$ can be detected. Regarding the toluene oxidation, 24 h-ageing promoted the performance of $Co_3O_4$ which can be associated with its larger SSA. However, this treatment was particularly unfavourable for propane oxidation: $T_{90}$ increased from 238 to 282 °C.

This fact could be associated with the different reaction mechanism or rate-determining step observed in the oxidation of toluene and propane [33]. Regarding the 80 °C-precipitating catalyst, it performed worse in the oxidation of toluene and propane than the room temperature-precipitating $Co_3O_4$, suggesting that heating treatment is unnecessary for its precipitation.

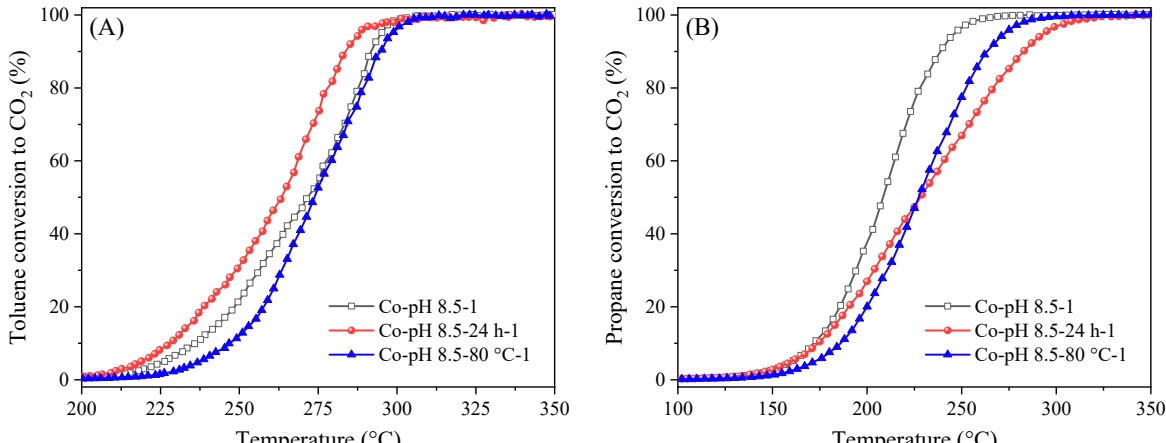

**Figure 11.** (**A**) Toluene and (**B**) propane oxidation over cobalt oxides prepared at different conditions.

**Table 5.** Temperature values of 10% ($T_{10}$), 50% ($T_{50}$) and 90% ($T_{90}$) toluene and propane conversion of cobalt oxides prepared at different conditions.

| Catalysts | Toluene Oxidation | | | Propane Oxidation | | |
|---|---|---|---|---|---|---|
| | $T_{10}$ | $T_{50}$ | $T_{90}$ | $T_{10}$ | $T_{50}$ | $T_{90}$ |
| $CoNH_3$-8.5-1 | 236 | 271 | 291 | 171 | 208 | 238 |
| $CoNH_3$-8.5-24 h-1 | 228 | 263 | 283 | 173 | 228 | 282 |
| $CoNH_3$-8.5-80 °C-1 | 246 | 273 | 294 | 186 | 227 | 263 |

## 3. Experimental Section

### 3.1. Materials

Ammonia aqueous ($NH_4OH$, ACS reagent, 25% $NH_3$ basis), cobalt(II) nitrate hexahydrate ($Co(NO_3)_2 \cdot 6H_2O$, reagent grade, 98%) and toluene (ACS reagent, 99.7%) were purchased from Sigma-Aldrich company (St. Louis, MO, USA) and were used as received without further treatment.

### 3.2. Catalysts Preparation

A total of 50 mL of $NH_4OH$ aqueous solution (V = 5 mL) was added drop by drop to 150 mL of $Co(NO_3)_2 \cdot 6H_2O$ aqueous solution (m = 5.8206 g), under vigorous stirring at room temperature. Once $NH_4OH$ aqueous solution was added, green precipitate was generated. The pH of the mixture was first adjusted with $NH_4OH$ aqueous solution to 8.0, 8.5, 9.0, 9.5 and 10.0, and next online monitored by a portable Radiometer analytical PHM201 pH meter (Hach, Loveland, Colorado, USA) and continuously maintained at the corresponding value for 1 h using $NH_4OH$ aqueous solution. The resultant precipitate was separated by centrifugation at a speed of ca. 2500 r/min for 5 min. The green precipitate corresponded to hydrated cobalt hydroxide while the light pink colour of the supernatant was due to the presence of cobalt ions. No colour difference was observed in the solids precipitated at different pH values. After being dried at 80 °C overnight, the solids were calcined in a furnace under static air at 200 °C for 2 h and then at 500 °C for 2 h (2 °C min$^{-1}$). The collected black powder was weighted in order to estimate the product yield. The obtained samples were named as Co-x, where x represented the precipitating pH values.

Another batch of samples was synthesized by a similar procedure by fixing the precipitating pH at 8.5 while varying the number of washings of the precipitant. The volume of washing water considered in each washing step was ca. 180 mL. The obtained samples were named to as Co-pH8.5-y, where y represented washing times.

In some cases, the precipitating pH was fixed at 8.5 and the precipitate was washed once whereas different ageing time (24 h) and precipitating temperature (80 °C) were set during the precipitation process; the corresponding samples were named to as Co-pH 8.5-24 h-1 and Co-pH 8.5-80 °C-1.

### 3.3. Catalysts Characterization

TG/DTA were carried out over the 80 °C-dried cobalt precursors from 25 °C to 625 °C (10 °C min$^{-1}$) on a SETARAM Setsys Evolution 12 calorimeter (SETARAM, Caluire, France), using 6–10 mg of samples under flowing air (50 mL min$^{-1}$). An empty 70-mL aluminium pan was used as the blank reference.

Nitrogen adsorption−desorption isotherms were obtained using a TRISTAR II apparatus (Micromeritics, Norcross, GA, USA) at −196 °C. Before analysis, each sample was degassed at 300 °C for 3 h. The specific surface areas were determined by the standard Brunauer–Emmett–Teller (BET) method. The total pore volume and the pore size distribution were calculated using the Barrett–Joyner–Halenda (BJH) method.

XRD patterns were recorded at room temperature in Bragg–Brentano parafocusing geometry using a D8 advance A25 diffractometer (Bruker, Karlsruhe, Germany) equipped with a Cu K$\alpha$ 1+2 radiation ($\lambda$ = 0.154184 nm) and a graphite monochromator on the diffracted beam. Samples were scanned from 10° < 2θ < 80° with a step size of 0.02° and a counting time of 2 s per step.

FTIR spectra were recorded on a FT-IR C92712 spectrometer (PerkinElmer, Waltham, MA, USA) in attenuated total reflectance mode at an instrument resolution of 4 cm$^{-1}$ over a range of 400–4000 cm$^{-1}$.

### 3.4. Catalytic Performance Evaluation

For each test, 150 mg of catalyst mixed with 700 mg of silicon carbide was packed inside a U-shaped reactor (220 mm in length and 4 mm in internal diameter), with a bed height of 6 mm. Silicon carbide was used in order to minimize the effect of hot spots.

#### 3.4.1. Complete Oxidation of Toluene

For toluene oxidation tests, the reactant gas mixture, composed of 1000 ppm toluene and synthetic air (21 vol.% O$_2$+79 vol.% N$_2$), with a total volumetric gas flow of 100 mL min$^{-1}$, was fed into the reactor before being heated from room temperature to 150 °C (5 °C min$^{-1}$) and held for 0.5 h in order to stabilize the system. Then, a second temperature ramp of 2 °C min$^{-1}$ was run until 350 °C and held for 1 h. Next, the reactor was cooled down to 150 °C (2 °C min$^{-1}$). The temperature of the catalyst bed was measured using a thermocouple. The concentrations of CO and CO$_2$ were in-situ recorded by a Rosemount X-stream Gas Infrared Analyzer (Emerson Electric Co., St. Louis, MO, USA). The toluene conversion was calculated as follows:

$$X_{C_7H_8}(\%) = \frac{[CO_2]}{7[C_7H_8]} \times 100$$

where [CO$_2$] and [C$_7$H$_8$] represent the outlet CO$_2$ concentration and the initial toluene concentration, respectively.

#### 3.4.2. Complete Oxidation of Propane

Regarding propane oxidation tests, after 100 mL min$^{-1}$ of the reactant gas mixture (0.1 vol.% propane + 21 vol.% O$_2$+79 vol.% He) was introduced into the reactor at room temperature, the reactor was heated from room temperature to 100 °C (5 °C min$^{-1}$) and held for 0.5 h in order to stabilize the system. Subsequently, the temperature was increased from 100 °C to 350 °C (2 °C min$^{-1}$) and held for

1 h. Next, the reactor was cooled down to 100 °C (2 °C min$^{-1}$). Gas effluents were analysed by an on-line micro gas chromatograph (SRA, Lyon, France) equipped with a thermal conductivity detector. The propane conversion was calculated as follows:

$$X_{C_3H_8}(\%) = \frac{[CO_2]}{3[C_3H_8]} \times 100$$

where $[CO_2]$ and $[C_3H_8]$ are the outlet $CO_2$ concentration and the initial propane concentration, respectively.

## 4. Conclusions

$Co_3O_4$-based catalysts prepared via ammonia-precipitation were synthesized, characterized, and tested for the total oxidation of toluene and propane. The effect of the precipitating pH and number of washings on the yields, physicochemical properties, and catalytic activity of $Co_3O_4$ was investigated. The results showed that precipitating $Co^{2+}$ between pH 8.5 and pH 10 can lead to $Co_3O_4$ yields above 87%, with the highest yield (92%) achieved at pH 9.0. Samples precipitated at pH 9 and 8.5 were shown to have the best catalytic performance for the oxidation of toluene and propane, respectively. As revealed by TG/DTA and FTIR, impurities existed in cobalt precursor prepared without the washing step, resulting in poor catalytic activity. Washing could improve the performance of the catalyst, especially for the sample washed twice ($T_{50}$ of 268 °C and 213 °C for toluene and propane oxidation, respectively) although negatively affected the $Co_3O_4$ yield. Moreover, long-term ageing and high-temperature heating during precipitation were proven to be unnecessary because they also negatively impact the catalytic performance.

**Supplementary Materials:** The following are available online at http://www.mdpi.com/2073-4344/10/8/900/s1, Figure S1: Variation of (A) toluene conversion (B) propane conversion to $CO_2$ with the reaction temperature during three consecutive cooling cycles over Co-pH 9.0 catalyst, Figure S2: (A) $N_2$ adsorption−desorption isotherms and (B) pore size distributions of cobalt oxides prepared at different conditions, Table S1: Product yields and textural data of cobalt oxides prepared at different conditions.

**Author Contributions:** This work was finished in collaboration with all authors. I.D. and W.Z. have prepared the catalysts, performed structural characterizations and catalytic oxidation tests, and written the manuscript. W.Z., J.L.V., M.H. and A.G.-F. coordinated the whole study, including data interpretation, result discussion, and manuscript review and revision. All authors have read and agreed to the published version of the manuscript.

**Funding:** This research received no external funding.

**Acknowledgments:** This work was financially supported by the University Claude Bernard Lyon 1, the CNRS and the Auvergne Rhone Alpes Region (project PAI 2019 LS 203067). We gratefully acknowledge the China Scholarship Council of China for Weidong Zhang grant and the IDEXLYON Starmac program for Imane Driouch'grant.

**Conflicts of Interest:** The authors declare no conflicts of interest.

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
