# Peer review of "Total Oxidation of Toluene and Propane over Co3O4 Catalysts: Influence of Precipitating pH and Washing"

_catalysts, doi:10.3390/catal10080900_

Round 1

Reviewer 1 Report

After reading the article titled total oxidation of toluene and propane over Co3O4 catalyst: Influence of precipitating pH and washing I would like to make some comments to the authors.

First, I have found some typographical errors on line 58 and 84.

In general, in my opinion, I see a descriptive article, where the authors have not discussed the results or have been very deep conclusions. The reader has many doubts after reading it. For example, because catalysts have good activity for the oxidation of toluene and not for propane. What exactly happens with the washes or the real influence of pH on the synthesis. The authors only suggest conclusions. As the title of the article influences pH and washes, the authors should study these aspects in-depth and they should give explanations and data for their conclusions. Perhaps, some additional techniques such as XPS to observe the chemical composition or some SEM photographs to observe the morphology can help to show the processes that occur, they could find a structure-activity relationship that is impossible to find with the data provided in the article.

Then, the first study of the influence of pH, in Table 1, no relationship is found between the pH and the structure, making it unclear what the influence of pH is on the synthesis. But after reading the article, when discussing the influence of the washes on the catalyst, the reader can see that after precipitation, many impurities remain in the catalyst, these impurities cannot be controlled, leaving a random amount in each catalyst synthesis. This may be the reason why the authors do not find any relationship in the influence of pH, since, in each solid, an undetermined amount of impurities remain, it affects each synthesized materials differently. This fact is confirmed in the ATG (figure 1A), where each solid loses a different amount of weight, and the Co-pH 8.0 catalyst line is also missing.

The authors should also discuss figure 6B more, not just describe it, for example, why exothermic peaks disappear and an endothermic peak appears and after, it disappears with washing.

I would like to know the opinion of the authors on the above comments.

Reviewer 2 Report

The manuscript presents the effects of precipitation pH and the number of washings on the structure of bulk cobalt oxide catalysts for total oxidation of toluene and propane at temperatures less than 350 C. A series of bulk cobalt oxide catalysts were synthesized via ammonia precipitation at different pH ranging from 8 to 10 and characterized using FTIR, XRD, SSA, and TGA techniques. It was shown that the washed samples prepared at pH 8.5 and 9 performed better among others, more washing was not helpful, and the total improvement was not significant. The following comments should be addressed before considering for publication in Catalysts.

  1. The manuscript has typos (especially question mark) and must be fully revised before publication.
  2. The errors of BET surface area measurements should be reported, authors reported the BET values with decimal values and made a conclusion based on them which is not reliable.
  3. The catalysts precursors and all utilized materials should be reported in the Experimental section (as a separate sub-section called Materials) along with their grade, purity, and source.
  4. The experimental part on the preparation of catalysts should be revised and more detailed information such as centrifugation time and speed, the way that pH was controlled, the color changes during precipitation should be added to the manuscript.
  5. It would also be nice if authors could include the TEM images of some of their synthesized samples to the manuscript to show the shape of nanoparticles and compare their sizes.

Reviewer 3 Report

This paper is a complete study on a wide number of preparation variables for the synthesis of bulk Co3O4 for the combustion of short-chain hydrocarbons. This is a very interesting work. However, to make a more complete contribution some characterisation data related to redox properties and Co3+/Co2+ surface molar ratio. Likewise, some kinetic results would be also welcome.

Round 2

Reviewer 1 Report

I am agree with the authors comment.  In my opinion the article is suitable for publication in catalyst.